# Enhancing quantum sensing sensitivity by a quantum memory

Sebastian Zaiser[1], Torsten Rendler[1], Ingmar Jakobi[1], Thomas Wolf[1], Sang-Yun Lee[1], Samuel Wagner[1], Ville Bergholm[2], Thomas Schulte-Herbrüggen[2], Philipp Neumann[1] & Jörg Wrachtrup[1]

In quantum sensing, precision is typically limited by the maximum time interval over which phase can be accumulated. Memories have been used to enhance this time interval beyond the coherence lifetime and thus gain precision. Here, we demonstrate that by using a quantum memory an increased sensitivity can also be achieved. To this end, we use entanglement in a hybrid spin system comprising a sensing and a memory qubit associated with a single nitrogen-vacancy centre in diamond. With the memory we retain the full quantum state even after coherence decay of the sensor, which enables coherent interaction with distinct weakly coupled nuclear spin qubits. We benchmark the performance of our hybrid quantum system against use of the sensing qubit alone by gradually increasing the entanglement of sensor and memory. We further apply this quantum sensor-memory pair for high-resolution NMR spectroscopy of single $^{13}C$ nuclear spins.

[1] 3rd Physics Institute, University of Stuttgart, Pfaffenwaldring 57, Stuttgart 70569, Germany. [2] Department of Chemistry, Technical University Munich, Lichtenbergstrasse 4, Garching 85747, Germany. Correspondence and requests for materials should be addressed to P.N. (email: p.neumann@physik.uni-stuttgart.de).

Precision sensing using quantum states usually relies on accurate measurements of their quantum phase. However, numerous susceptibilities to environmental noise make quantum states fragile resulting in limited sensitivity. Therefore, the acquisition of a large phase is a central challenge[1–5]. Typically, two strategies are used to enhance quantum sensing: one way is multi-particle entanglement of sensing qubits which results in rapid phase accumulation, but often is counterbalanced by faster dephasing[6–8]. Entanglement, however, pays off, if fluctuations of the quantity to be measured have a shorter correlation time than the single sensor coherence time. A long-lived memory is another way which is advantageous when the quantity to be measured has a longer correlation time than the sensor's coherence time[9–12]. The quantum memory approach is particularly suited in hybrid sensor systems where the sensing qubit strongly interacts with the quantity to be measured while the storage qubit is well isolated from environmental influences except for its coupling to the sensor. Typically, highest sensitivity is reached when the available coherence time of the sensing qubit is most effectively used[13–15]. Further enhancement is gained if the quantum state of the sensing qubit is at least partially stored for a later feedback, and thereby exploiting the observable's longer correlation time[9,10,12].

Here, we utilize a quantum memory for full storage of the sensor's quantum state leading to enhanced sensitivity. We demonstrate that entanglement of quantum memory and sensing qubit during the phase accumulation process makes efficient use of the resources at hand. The enhanced sensing time due to use of the quantum memory improves spectral resolution. A further benefit of our phase estimation-type[16] protocol are non-local gate operations between quantum memory and, for example, a sample spin.

We exploit the benefits of the quantum memory by a measurement protocol based on the correlating two subsequent phase estimation steps. If the correlation time of the measured quantity (for example, a magnetic field) is longer than the coherence time of the sensor, this coherence time limits the measurement resolution and makes recording of the dynamics challenging. To recover dynamics and increase spectral resolution, subsequently measured quantities $S$ can be correlated (that is, $\langle S(t)S(t+\tau)\rangle$). However, because the measurement exhibits a limited visibility $A<1$ of the measurement signal $S$, the visibility of a correlation measurement decreases as $\propto A^2$. One solution are classical correlation measurements[9,10,17–23], where a long-lived memory stores information. If memory and sensing qubit are not entangled, typically half of the signal amplitude $A$ is lost. Full SWAP-gates between quantum sensor and quantum memory on the other hand require excessive operation on the quantum register, which leads to a reduced overall fidelity. Here, we retain the full measurement contrast $A$ by entangling sensor and memory qubit. Furthermore, we show that storing the full quantum state not only allows for improved detection of weakly coupled qubits, but also enables coherent interaction and non-local gates between memory and distinct weakly coupled qubits.

## Results

**Entangling sensor and memory qubit.** We use the electron spin of a nitrogen-vacancy (NV) centre in diamond and the associated $^{14}$N nuclear spin[14] to build a hybrid quantum sensor-memory pair (see Fig. 1a,b). While the electron spin serves as a magnetic field sensor, the nuclear spin acts as a quantum memory due to its much weaker coupling to the environment. The coherence of the memory decays slightly slower than the electron spin lattice relaxation time[14], $T_{1,\text{sensor}} = (4.94 \pm 0.16)$ ms, whereas the sensor coherence decays more than one order of magnitude faster in a spin echo measurement, $T_{2,\text{sensor}} = (395 \pm 5)\,\mu$s (see Fig. 1d). Hence, we can gain up to one order of magnitude in frequency and equivalently field resolution by using the nuclear spin as

memory. To demonstrate the role of entanglement between the sensor and the quantum memory we gradually increase entanglement between them and measure the resulting sensing accuracy. We further test the novel scheme for entangling the memory qubit and a proximal $^{13}$C nuclear spin. We use the tiny magnetic field of weakly coupled $^{13}$C nuclear spins as measurement quantity with extremely long correlation time.

The algorithm we apply to make efficient use of the sensing qubit and the quantum memory resembles a phase estimation procedure. Example metrology devices based on the phase estimation are atomic clocks[1], vapour-cell magnetometers[2] or nanoscopic spin-based sensors such as the NV centre[3,4], just to name a few. In essence, we perform two phase accumulation steps separated by the correlation time $T_c$ with one sensing/processing qubit and one storage qubit as shown by the quantum circuits in Fig. 1c (for example, see (ref. 16)). Each phase accumulation step comprises a pair of entangling and disentangling gates separated by time $\tau$. The first gate in each step entangles sensor and memory qubits. The degree of entanglement is expressed by the variable $\eta$ ($\eta = 0$ for no entanglement, $\eta = 1$ for a Bell state), which relates to the entanglement measure negativity $\mathcal{N}$[24,25] (see the 'Methods' section) as $\mathcal{N} = \sin(\eta\pi/2)/2$. During $T_c$ the phase, which is (mostly) acquired by the sensing qubit, remains stored in the memory. As $\mathcal{N}$ increases up to 0.5, the full quantum information (that is, the full complex phase factor $e^{i\phi}$) is stored on the memory. For no entanglement (that is, $\mathcal{N}=0$), however, only half of the information is stored on the memory (for example, $\cos\phi$). The two cases of full and no entanglement will be referred to as enhanced and conventional sequence, respectively (see Fig. 1c and see the 'Methods' section).

Next, we describe the enhanced sequence. As an initial step, we prepare the electron (sensor) and the $^{14}$N (memory) spins in a superposition state (for basis states, see Fig. 1b).

$$|\Psi_i\rangle = |0\rangle \otimes (|0\rangle + |1\rangle) \tag{1}$$

This quantum state essentially does not acquire any phase because of the weak nuclear magnetic moment. Next, we prepare a fully entangled state by applying a nuclear-spin-controlled rotation of the electron spin ($C_n\text{ROT}_e$-gate, equation 2).

$$|\Psi_0\rangle = |00\rangle + |11\rangle \tag{2}$$

During a free evolution time $\tau$, a phase $\phi_1$ is accumulated leading to $|00\rangle + e^{i\phi_1}|11\rangle$ with $\phi_1 = -\gamma_e B_z\tau$ (gyromagnetic ratio of the electron spin $\gamma_e$, magnetic field $z$-component $B_z$). The subsequently applied $C_n\text{ROT}_e$-gate disentangles the sensor and memory spins (equation 3). Now, the accumulated phase is stored in the memory, while no sensing information remains on the sensing qubit.

$$|\Psi_1\rangle = |1\rangle \otimes (|0\rangle + e^{i\phi_1}|1\rangle) \tag{3}$$

Note, that the first and second $C_n\text{ROT}_e$-gate are controlled by the $|1\rangle$- and $|0\rangle$-state, respectively. Therefore, the electron spin state is $|1\rangle$ after the second gate.

During the following correlation time $T_c$, changes in external parameters like magnetic field do not change the stored phase, except for a potential detuning $\Delta\omega$ (equation 4 and see the 'Methods' section).

$$|\Psi_2\rangle = |1\rangle \otimes (|0\rangle + e^{i(\phi_1 + \Delta\omega T_c)}|1\rangle) \tag{4}$$

Please note, that a detuning does not affect the classical memory. After the correlation time $T_c$ we repeat the pair of $C_n\text{ROT}_e$-gates and find our spin system in state

$$|\Psi_f\rangle = |0\rangle \otimes (|0\rangle + e^{i(\Delta\phi + \Delta\omega T_c)}|1\rangle). \tag{5}$$

The particular order of $|1\rangle$- and $|0\rangle$-controlled $C_n\text{ROT}_e$-gates ultimately refocuses any quasi static magnetic fields. Any change

in local magnetic field during $T_c$, however, leads to a phase difference $\Delta\phi = \phi_1 - \phi_2$ after the second $\tau$ period, which remains on the memory qubit and constitutes our metrology information.

The phase of the memory qubit is finally read out as probabilities $S_{enh}$ and $S_{conv}$ to find the memory qubit in state $|1\rangle$[14,26],

$$S_{enh} = \frac{1+c}{2} - A\frac{\cos(\Delta\omega T_c + \Delta\phi - \varphi)}{2} \tag{6}$$

$$S_{conv} = \frac{1+c}{2} - A\frac{\langle\cos^2\phi - \cos\phi\sin\phi\rangle\cos(\Delta\phi - \varphi)}{2}$$
$$= \frac{1+c}{2} - \frac{A}{2}\frac{\cos(\Delta\phi - \varphi)}{2} \tag{7}$$

for enhanced and conventional sequence (see Fig. 1 and see the 'Methods' section). In the conventional sequence, the decay of

residual coherence on the sensor after the first $\tau$ enters by averaging over a range of $\phi$ larger than $2\pi$, which leads to an additional factor 1/2 in equation 7. Hence, the signal contrast $A_{enh} = A$ for the enhanced sequence is twice as large as the contrast $A_{conv} = A/2$ for the conventional sequence. Imperfect initialization and readout of the sensor and memory system result in finite signal contrast $A < 1$ and a signal shift $c \approx 0.2$ ($0 \leq S \leq 1$) (ref. 27; see see the 'Methods' section).

The described enhanced sensing sequence is very efficient regarding the number of quantum gates, energy deposited on the sample and time. Our entanglement-based approach facilitates direct quantum memory access for the sensor, where only $C_nROT_e$-gates and no time- and energy-intensive SWAP-gates are required. The duration of the $C_nROT_e$-gates is limited by the hyperfine coupling between sensor and memory (that is,

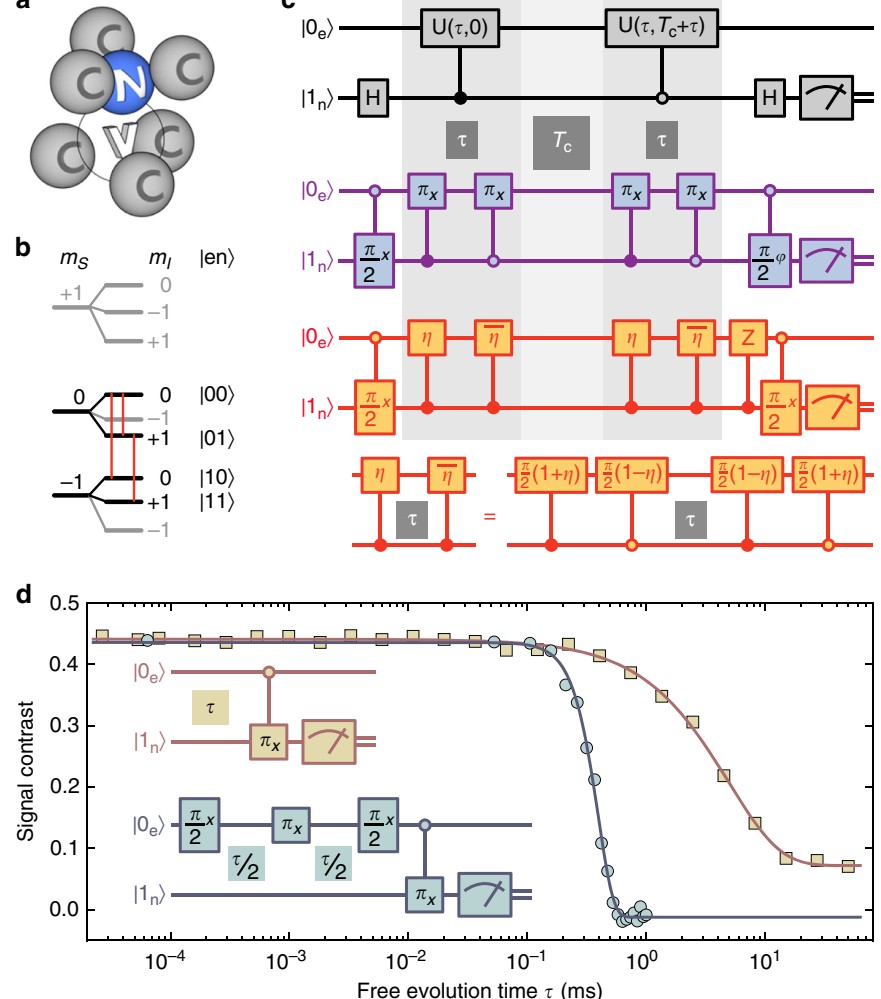

**Figure 1 | System parameters and measurement sequence.** (**a**) Structure of an NV defect centre in diamond[45]. Two adjacent lattice sites are occupied by a nitrogen atom and a vacancy. (**b**) Energy-level structure of NV electron spin e ($S = 1$) and hyperfine coupled $^{14}$N nuclear spin n ($I = 1$). Four out of nine spin states form the two-qubit basis-states $|en\rangle$. They are eigenstates to electron spin operator $S_z$ and nuclear spin operator $I_z$ with respective magnetic quantum numbers $m_S$ and $m_I$ (see the 'Methods' section). Addressed spin transitions are marked with vertical red lines. (**c**) Quantum wire diagrams for quantum phase correlation measurements. Grey indicates general idea. Two quantum gates $U(\tau, t)$ controlled by the memory qubit state represent our phase estimation steps with duration $\tau$ at times $t$ (that is, with separation $T_c$). Filled (open) circles indicate $|1\rangle - (|0\rangle - )$ controlled gates. If both quantum gates are equal, the total phase on the memory is zero. Purple indicates the enhanced sequence with full entanglement of sensor and memory. Experimental implementation using controlled spin rotations (for example, $\pi/2$-pulses and $\pi$-pulses) of the sensor (memory) controlled by the memory (sensor) and free evolution times $\tau$ and $T_c$. Orange indicates sequences with variable degree of entanglement between sensor and memory realized via the $\eta$- and $\bar{\eta}$-gate (see the 'Methods' section). (**d**) Measurements of the electron spin's $T_1$ and $T_2$ time via measurement of the spin state population decay and a spin echo measurement (squares and circles). The inset shows the quantum wire diagrams of the respective measurement sequences. The signal is the probability of detecting the memory spin state $|1\rangle$ and the contrast is the difference of two signals with the final $\pi$-pulse on the memory controlled by the $|0\rangle$ and $|1\rangle$ state of the sensor.

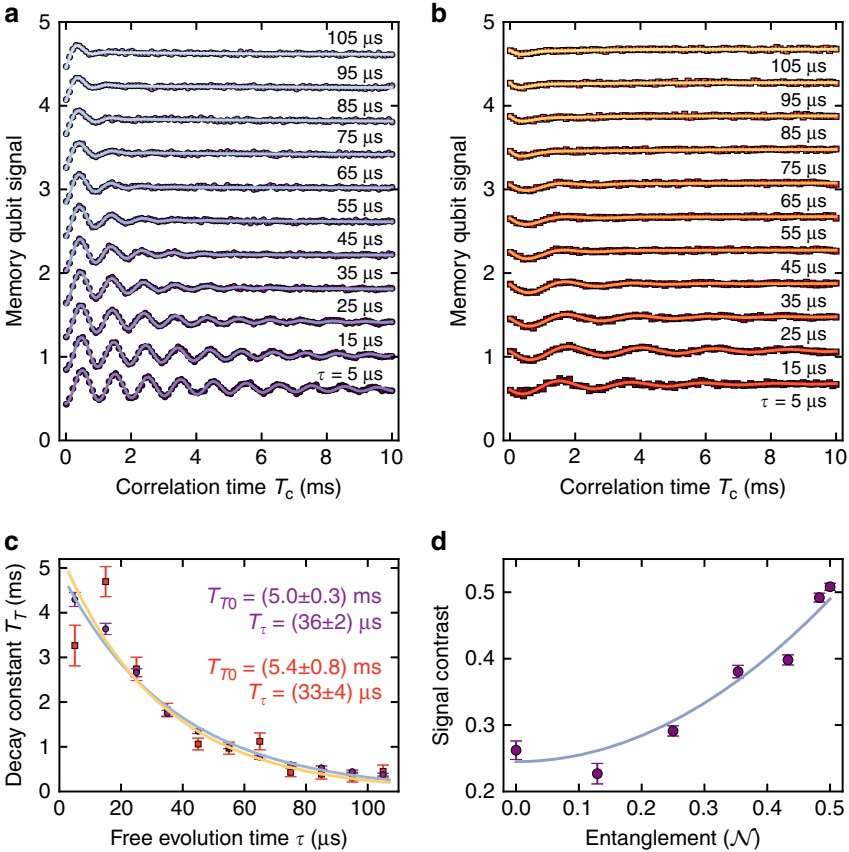

**Figure 2 | Comparison of enhanced and conventional correlation measurements.** (**a,b**) Stacks of correlation measurements for full ((**a**) $\eta = 1$) and no entanglement ((**b**) $\eta = 0$) between sensor and memory qubit for various sensing times $\tau$ as stacking parameter. Associated measurement sequences are given in Fig. 1c. The memory readout signal (that is, the probability of detecting state $|1\rangle$) is plotted against the correlation time $T_c$ (stack-offset $n \cdot 0.4$, $n \in \{0, 1, \ldots 10\}$). Oscillations on the signal are introduced artificially by a phase offset $\varphi = 2\pi \cdot T_c \cdot f_T$, $f_T = 1 \, \text{kHz}$ (**a**) and $f_T = 0.5 \, \text{kHz}$ (**b**) to sample the stored quantum phase (see equations 6 and 7). All data sets are fit by $y = y_0 + (A_0/2)\exp\{-T_c/T_T\}\cos(\varphi + \varphi_0) + A_1 \exp\{-T_c/T_1\}$. (**c**) Fit values for decay constant $T_T$ over $\tau$ for no (squares, orange) and full (circles, purple) entanglement of sensor and memory. The function $T_T = T_{T0} \exp\{-\tau/T_\tau\}$ is fit to the weighted data and the corresponding parameters are displayed. (**d**) Signal contrast $A_0$ for $T_c = 0$ over entanglement between sensor and memory qubit (given as negativity $\mathcal{N}$, see the 'Methods' section). All error bars give standard errors taken from the used least square fit.

$A_{zz} = -2.16 \, \text{MHz}$, cf. equation 11). Here, the gate duration is 2 µs (see the 'Methods' section), which renders our enhanced sequence in total 8 µs longer than the conventional sequence (see the 'Methods' section and (refs 9–11) compared with a total sequence duration of a few milliseconds (see the 'Methods' section). In contrast, control of the nuclear spin memory (see the 'Methods' section) requires orders of magnitude larger radio-frequency (RF) fields than the electron spin sensor for similar quantum gates. For example, an alternative sequence utilizing SWAP-gates would take 275 µs longer (see the 'Methods' section).

Next, we compare the performance of enhanced and conventional sequence (Fig. 2a,b). As a figure of merit, we measure the phase information stored in the memory. According to equations 6 and 7 we express the latter quantity by the signal contrasts $A_{\text{enh}}$ and $A_{\text{conv}}$ where the signal is the probability to find the memory qubit in state $|1\rangle$ (see the 'Methods' section). Figure 2a,b show the memory signal for various sensing times $\tau$ and correlation times $T_c$. We visualize the stored phase $\Delta\phi$ on the memory qubit by increasing the phase $\varphi$ of the final memory $\pi/2$-pulse (Fig. 1c bottom) proportional to $T_c$ for both, enhanced and conventional measurement, respectively (see Fig. 2a,b). The resulting oscillating signals have contrasts $A_{\text{enh}}$ and $A_{\text{conv}}$ and decay with time constants $T_T$. The available correlation time $T_T$ is around $T_{T0} \approx 5 \, \text{ms} \approx T_{1,\text{sensor}}$ for short sensing times $\tau \approx 5 \, \mu\text{s}$ and

decays exponentially for increasing $\tau$ with a time constant $T_\tau \approx 35 \, \mu\text{s}$ (Fig. 2c). From Fig. 2a,b we can extract the amount of phase information $A_0$ for zero correlation time $T_c$ but varying sensing time $\tau$ (see the 'Methods' section). The $A_0$ values behave analogously to the signal of a spin echo (cf. Fig. 1d) and hence they do not decay on the probed $\tau$ range, which is below $T_{2,\text{sensor}}/2$. The $A_0$ values for the enhanced sequence are about twice as large as those of the conventional sequence. Summarizing, entangling the sensor and the memory qubit does not deteriorate the relaxation times intrinsic to the sensor qubit and yields the benefit of a doubled signal (see the 'Methods' section). In Fig. 2d we plot the phase information $A_0$ for varying entanglement $\mathcal{N}$ (see the 'Methods' section). Apparently, the stored information increases monotonously with increasing entanglement, as expected from equations 6 and 7.

**Quantum correlations between memory qubit and sample spin.** To demonstrate enhanced sensing by the quantum memory, we use our measurement sequence to establish quantum correlations between memory and a proximal $^{13}\text{C}$ nuclear spin with a hyperfine coupling strength $A_{zz} \approx 89 \, \text{kHz}$ to the electron spin. In addition, $^{13}\text{C}$ spins have very long $T_1$ relaxation times and therefore the corresponding magnetic field at the sensor position shows a particularly long correlation time with a time constant beyond seconds. In previous work we have demonstrated

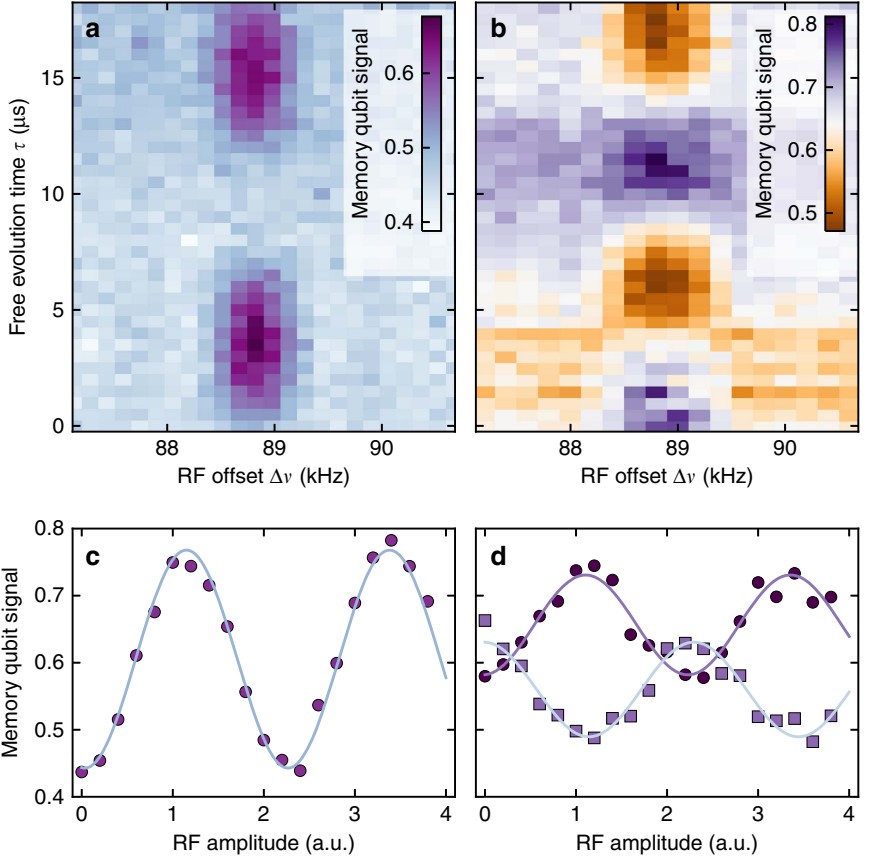

**Figure 3 | Sensing a single nuclear spin.** (**a,b**) Colour-coded memory qubit signals (that is, probability of detecting state $|1\rangle$) of enhanced correlation measurement with an RF pulse during $T_c = 1$ ms. Sensing time $\tau$ and RF pulse frequency offset $\Delta v$ to the bare $^{13}$C Larmor frequency are varied (cf. Mims electron-nuclear-double-resonance (ENDOR)[29]). A resonance of a $^{13}$C spin around $\Delta v = 89$ kHz appears, where, for increasing $\tau$, the phase $\Delta\phi$ of the memory increases. The width of the $^{13}$C spin controlled phase accumulation of about 1 kHz is governed by the RF pulse duration. **a,b** show $\cos\Delta\phi$ and $\sin\Delta\phi$ of the memory phase because the phase $\varphi$ of the final memory qubit $\pi/2$-pulse is set to 0 (**a**) and $\pi/2$ (**b**) (see Fig. 1c). Please note, that for the measurements done with $\varphi = \pi/2$ (**b**) initialization of the $^{13}$C spin is required (here: spin down, see main text). Variations of the background in panel (**b**) are due to drifts of $\Delta\omega$ (see equation 6, see the 'Methods' section). (**c,d**) On resonance, the RF pulse amplitude is varied, to induce Rabi oscillations of the detected $^{13}$C spin. Phases $\varphi$ are 0 in (**c**) and $\pm\pi/2$ in (**d**) for circles and squares.

initialization, coherent control and readout of such spins[28] which therefore serve as well-characterized sample quantum systems and deterministic sources of magnetic fields.

In the first sensing time $\tau$ of the measurement sequence we correlate the memory qubit's phase $\phi_1$ with the $^{13}$C spin's $z$-projection $m_I$ as $\phi_1 = 2\pi m_I A_{zz}\tau$, where $A_{zz}$ is the detuning due to hyperfine interaction (see below and see the 'Methods' section). Then, during the correlation time $T_c$, a resonant RF pulse of frequency $v$ flips the $^{13}$C by an angle $\alpha$[29]. For $\alpha = 0$, both spins are uncorrelated after the second sensing time. For $\alpha = \pi$, however, the phase on the memory spin is

$$
\begin{aligned}
\Delta\phi &= -2\pi\Delta m_I A_{zz}\tau \\
&= \mp 2\pi A_{zz}\tau
\end{aligned} \tag{8}
$$

when the $^{13}$C spin projection undergoes a change $\Delta m_I = \pm 1$ during $T_c$. After tracing out sensor and sample spin, the memory spin state before the final $\pi/2$-rotation (see Fig. 1c) is

$$
\rho^{(n)}_{\uparrow\downarrow} = \frac{1}{2}\begin{bmatrix} 1 & \cos^2\frac{\alpha}{2} + e^{-i\Delta\phi}\sin^2\frac{\alpha}{2} \\ \cos^2\frac{\alpha}{2} + e^{i\Delta\phi}\sin^2\frac{\alpha}{2} & 1 \end{bmatrix} \tag{9}
$$

$$
\rho^{(n)}_{mix} = \frac{1}{2}\begin{bmatrix} 1 & \cos^2\frac{\alpha}{2} + \cos\Delta\phi\sin^2\frac{\alpha}{2} \\ \cos^2\frac{\alpha}{2} + \cos\Delta\phi\sin^2\frac{\alpha}{2} & 1 \end{bmatrix} \tag{10}
$$

for the sample spin initially pointing up, down or being in a fully mixed state ($\uparrow$, $\downarrow$, mix). Hence, for $\alpha = \pi$, $\Delta\phi$ has the maximum effect on the final state, and furthermore, if $\Delta\phi = \pi$ (that is, $\tau = (2A_{zz})^{-1}$) the final memory state is orthogonal to the initial superposition state. It should be noted that only for a sample spin initialized into states $\uparrow$ or $\downarrow$ and $\alpha = \pi$ a pure memory spin state is observable (equation 9), whereas the correlation with a mixed sample spin state may bring the memory qubit into a mixed state (equation 10 for $\Delta\phi = \pi/2$).

For the following measurements we have chosen $\alpha = \pi$. We have plotted the memory qubit readout signal as a function of $\tau$ and $\Delta v$ in Fig. 3a,b where $\Delta v$ is the RF offset to the bare Larmor frequency $v_L$ of $^{13}$C spins (that is, $\Delta v = v - v_L$). Both measurements were performed with a detuning $\Delta v$ around the hyperfine coupling $A_{zz} \approx 89$ kHz between $^{13}$C and electron spin. While the data in Fig. 3a was measured with $\varphi = 0$ of the final memory $\pi/2$-pulse (cf. Fig. 1c centre (purple)), the measurement in Fig. 3b shows results with $\varphi = \pi/2$. Thus, we have measured $\cos\Delta\phi(\tau)$ and $\sin\Delta\phi(\tau)$ of the memory's phase $\Delta\phi$, respectively (cf. equation 6). Consequently, we see the corresponding oscillations for increasing $\tau$ when the RF is in resonance. Furthermore, in the $\cos\phi$ case a changing field due to a flip $\Delta m_I$ of the $^{13}$C spin is detected disregarding the direction of the flip, and in the $\sin\phi$ case we can discriminate between the $^{13}$C spin flipping up or down (that is, the sign of $\Delta m_I$). For example, Fig. 3b shows a flip $\Delta m_I = 1$ (that is,

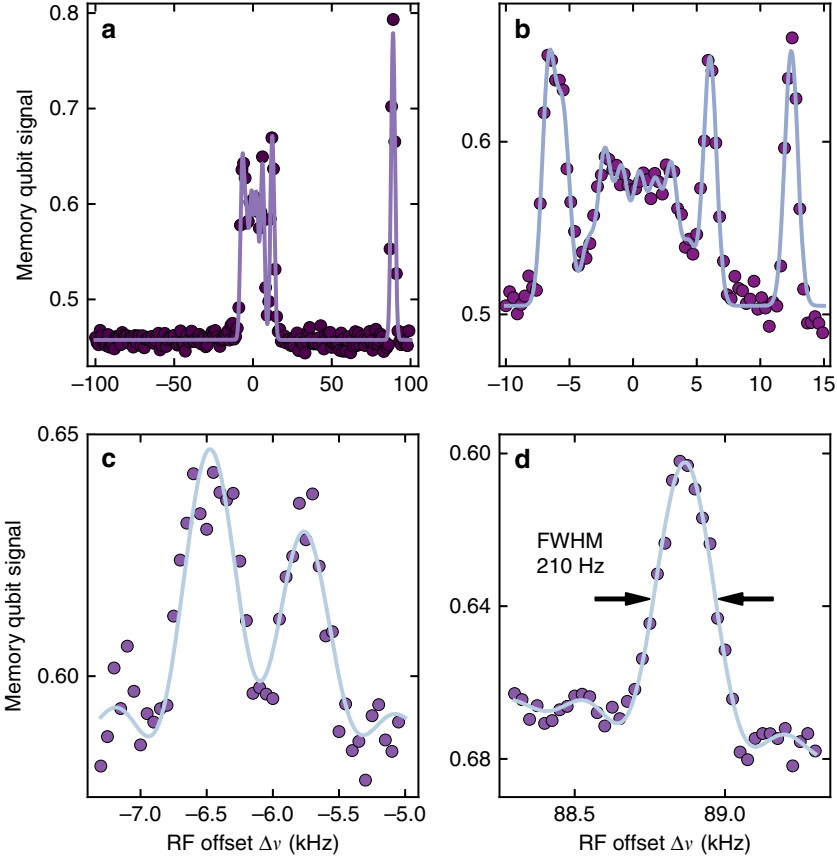

**Figure 4 | Resolving nuclear spins at the strong coupling limit.** (**a**–**d**) Enhanced Mims ENDOR spectra of hyperfine coupled $^{13}$C spins. During the entire correlation period $T_c$ the RF pulse is switched on. The duration of the correlation time $T_c$ is set to 0.5 ms (**a**) 1 ms (**b**) 2 ms (**c**) and 4.3 ms (**d**). The $^{13}$C RF pulses during $T_c$ were Gaussian (**a,b**) or rectangular (**c,d**), thus peaks were fit with Gaussian or sinc-functions, respectively. Spectral resolution increases with $T_c$. For the unresolved resonances within $-4...4$ kHz an empirically determined number of peaks is fit.

$\Delta\phi < 0$, cf. equation 8) as we have initialized the nuclear spin into the spin down state before the correlation sequence[28]. Note that the spectral window of coherent interaction in Fig. 3a,b has a width of ~1 kHz, compatible with the RF $\pi$ pulse of 1 ms duration, which is much longer than the sensor's coherence time $T_{2,sensor}$.

In the following, the RF was set on resonance (that is, $\Delta v \approx 89$ kHz) and the effective sensing time $\tau_{eff} = \tau_0 + \tau$ was chosen to be $\tau_{eff} = (2A_{zz})^{-1} \approx 5.5\,\mu$s and $\tau_{eff} = (4A_{zz})^{-1} \approx 2.8\,\mu$s yielding $\Delta\phi \approx \pi$ and $\pi/2$ for Fig. 3c and d, respectively (see 'Methods' section). Figure 3c,d show Rabi oscillations of the initialized $^{13}$C sample spin (that is, variations of $\alpha$ in equations 9 and 10) for $\varphi = 0$ and $\varphi = \pm\pi/2$, respectively. Assuming an intrinsic sensing time $\tau_0 \approx 2\,\mu$s during pairs of finite-duration $C_nROT_e$-gates (see the 'Methods' section), we have set $\tau$ to 3.5 and 0 $\mu$s respectively see 'Methods' section. Interestingly, for $\varphi = 0$ and RF amplitude around 0.5 ($\alpha = \pi/2$) the memory spin and the sample spin are entangled ($|0\downarrow\rangle - |1\uparrow\rangle$). Hence, our novel sensing sequence can establish coherent interactions among memory and sample spins with high spectral selectivity on the sample spins frequency.

**Enhanced high-resolution correlation spectroscopy.** In a final step, we alter the measurement sequence for the detection of weakly coupled $^{13}$C spins. Therefore, we sweep the RF offset $\Delta v$ of the RF pulses around the bare $^{13}$C Larmor frequency. In addition, we adjust the sensing time $\tau \approx (2A_{zz})^{-1}$ according to the expected hyperfine coupling $A_{zz} = \Delta v$ with a maximum $\tau = 40\,\mu$s. For a coarse spectrum we chose a correlation time of $T_c = 500\,\mu$s. The resulting spectrum in Fig. 4a reveals a $^{13}$C spin at 89 kHz (see also Fig. 3), and additional spins closer to zero hyperfine coupling. A higher spectral resolution is obtained in Fig. 4b where $T_c$ is set to 1 ms. When $T_c$ is set to 2 ms two formerly unresolved $^{13}$C spins at around $-6$ kHz coupling are now distinguishable (see Fig. 4c). The best spectral resolution of 210 Hz (FWHM) could be obtained for a correlation time of $T_c = 4.3$ ms (see Fig. 4d).

## Discussion

Summarizing, we have implemented an entangled hybrid quantum sensor-memory pair for improved correlation spectroscopy, which outperforms the sole sensor by a factor of two in measurement signal and hence a factor of four in measurement time to reach an identical signal-to-noise ratio. Furthermore, the addition of the memory did not show any noticeable disadvantages. Using the enhanced correlation spectroscopy method, we were able to detect, discriminate and coherently couple to distinct $^{13}$C nuclear spins with high spectral resolution. Our sensor-memory pair method is compatible with other recently developed classical correlation methods[9–11]. Our method, for example, might improve proposals where an ancillary nuclear spin memory should be utilized for improved spatial resolution of sample spins[12]. The demonstrated coherent interaction with weakly coupled qubits facilitated by the quantum memory is particularly interesting for preparation, steering and readout of larger scale quantum simulators as proposed in ref. 30. Most importantly, our approach is complementary to others exploiting entanglement, as for

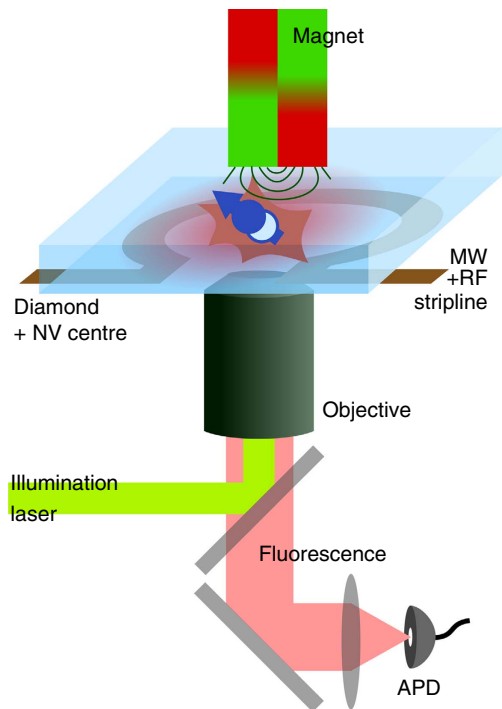

**Figure 5 | Experimental setup.** Confocal microscope illuminating and detecting the fluorescence response of a single NV centre in diamond. Spin levels are altered by an external magnetic field, which is here aligned with the NV symmetry axis represented by the arrow. Spin transitions are addressed via driving in the MW (electron spin) and RF (nuclear spins) regime. Both frequencies are delivered by the same microstructed MW + RF stripline antenna.

example GHZ states or spin-squeezed states of multiple sensors. In the first case, the quantum phase might be stored on a single memory as in our case, whereas in the latter case each sensor participating in the squeezed state needs a separate memory. An interesting test system for the latter would be an ensemble of NV centres[31], where each centre intrinsically comprises a sensor and a memory. For this system, the beneficial implementation of squeezing has been proposed[32]. Furthermore, the application of quantum-error-correction protocols[28,33,34] may extend the nuclear spin's storage time beyond the $T_1$ time of the electron spin. Another alternative would be the storage into weakly coupled memory spins which could potentially be completely decoupled from the electron spin[35] reaching storage times of seconds and corresponding spectral resolutions below one Hz. As a final remark, our correlation measurement resembles Kitaev's phase estimation algorithm using just a single-phase accumulation step. More efficient phase estimation can be achieved by increasing the number of memory qubits and performing a final inverse quantum Fourier transformation.

## Methods

**Experimental conditions.** *Experimental setup and used diamond.* The host diamond crystal, which was used in this study, is a type IIa CVD layer with [100] surface orientation and a $^{13}$C concentration of 0.2% (ref. 36). The used NV centre mainly appears negatively charged, that is, NV$^-$ (ref. 27). All mentioned experiments are performed in that negative charge state, if another charge state is not explicitly mentioned . We omit the '$-$' sign throughout the paper. It is located $\sim 15\,\mu$m below the diamond surface and a solid immersion lens has been carved around it via focused ion beam milling[37]. A coplanar waveguide for microwave (MW) and RF excitation made from copper is fabricated onto the diamond via optical lithography. An external static magnetic field of $\sim 540\,$mT from a permanent magnet is aligned along the symmetry axis of the NV centre (z-axis).

All the experiments were carried out at room temperature (see Fig. 5). The NV centre of interest is addressed in a homebuilt confocal microscope, where 532 nm light is focused onto the NV by an oil immersion objective, which also collects light from the diamond. The fluorescence light of the NV is isolated by spatial and spectral filtering and finally detected with a single photon counting detector[38].

Optical excitation of the NV centre on the one hand polarizes the electron spin triplet ($S = 1$) into its $m_S = 0$ state of the spin projection operator $S_z$. On the other hand, the photoluminescence intensity of the NV centre is correlated to the $m_S$ state, it is high for $m_S = 0$ and equally low for $m_S = \pm 1$ (ref. 38). The latter enables optically detected magnetic resonance of the electron spin. The correlation of electron spin state and the number of photoluminescence photons is too weak to gain knowledge about the current spin state before it decays or is re-initialized. This is different for certain nuclear spins at room temperature[26,39] and for the electron spin itself at cryogenic temperatures[40]. Indeed, here we apply the described high magnetic field aligned along the axis connecting nitrogen atom and vacancy, to ensure single-shot readout (SSR) capability for the nitrogen nuclear spin and at least the two strongest coupled $^{13}$C spins[28]. For more information about nuclear spin SSR see the according section below.

*Spin Hamiltonian.* The interaction of electron and nuclear spins is explained via the spin Hamiltonian

$$H = DS_z^2 - \gamma_e B_z S_z - \sum_i \gamma_{n,i} B_z I_{z,i} + \sum_i \underbrace{S\mathcal{A}_i I_i}_{\approx A_{zz,i} S_z I_{z,i}} \qquad (11)$$

Where, $D = 2.87\,$GHz is the zero-field splitting, $B_z$ is the $z$-component of the magnetic field, $\gamma_e \approx -28\,$GHz/T and $\gamma_n = 3.08\,$MHz/T and $10.71\,$MHz/T are the gyromagnetic ratios of the electron and $^{14}$N and $^{13}$C nuclear spins and $\mathcal{A}_i$ is the hyperfine coupling tensor for nuclear spin $i$. When the hyperfine contributions $A_{nm}$ of $\mathcal{A}_i$ are much smaller than the other parts of the Hamiltonian, the given approximation is justified, that is, only $A_{zz}$ does commute with the other parts of the Hamiltonian and its effect is therefore not suppressed. For $^{14}$N we have $A_{zz} = -2.16\,$MHz and for the two strongest coupled $^{13}$C spins we have $A_{zz} = 413\,$kHz and 89 kHz (ref. 28). For properly working SSR of a particular $^{13}$C spin the $A_{zx}$ and $A_{zy}$ components of its hyperfine tensor should be much smaller than the nuclear Zeeman term[26,39] and the $A_{zz}$ coupling should be sufficiently strong to facilitate $C_n ROT_e$-gates between electron and nuclear spin. SSR of nuclear spins is explained in a separate section below. In this process the nuclear spin is projected into an $m_I$ state with respect to the nuclear-spin operator $I_z$.

*SSR and initialization of nuclear spins.* The basic element of the SSR of a nuclear spin comprises a laser pulse, including fluorescence detection, and a MW pulse (see Fig. 6d). More precisely, the electron spin is optically read out and simultaneously initialized into the $m_S = 0$ state ($|0\rangle$). A subsequent $C_n ROT_e$-gate is synthesized by extensions of the optimal control platform DYNAMO[41]. It correlates the electron and nuclear spin projections $m_S$ and $m_I$ according to $m_S = 0 \mapsto m_I = 0, -1$ and $m_S = -1 \mapsto m_I = +1$. The electron spin readout of the subsequent SSR element yields partial information about the nuclear-spin state and this information is incremented by repeating the basic SSR element about a few hundred to thousand times, which takes up to a few milliseconds. Finally, the accumulated information comprises one binary SSR result (that is, '1' or '0') about the nuclear spin being in state $|1\rangle$ or not in state $|1\rangle$ (that is, the number of fluorescence photons is below or above a threshold value[26]). The properties of such measurements fulfil the condition for a strong quantum nondemolition measurement[26,42]. Hence, the nuclear-spin state before the readout is strongly correlated with the SSR result, if it was an eigenstate of the measurement operator (that is, $|m_I\rangle$ and $I_z$). Furthermore, the nuclear spin state after the readout is also strongly correlated to the SSR result. Therefore, we utilize a single SSR result (that is, 0 or 1) for two purposes, namely readout and initialization of the nuclear spin. Non-unity SSR fidelity reflects imperfect correlations.

The memory qubit signal mentioned throughout this report and displayed in Figs 1 to 4 arises from a correlation of two subsequent SSR results each concluding a single measurement sequence. Namely, it is the probability to find the memory spin in state $|1\rangle$ if it was also in state $|1\rangle$ in the previous readout step. Hence, if the memory is not in state $|1\rangle$ after a measurement run it counts as uninitialized (that is, state $m_I = 0$ or $-1$), the readout result of the subsequent measurement run will be discarded. Deterministic initialization of a nuclear spin can be achieved by swapping the initialized electron spin state onto the nuclear spin and subsequently confirming successful initialization with an SSR measurement[28].

The SSR signal has a visibility smaller than one due to a limited readout fidelity. Furthermore, the signal is not symmetric, around 0.5, but rather around 0.6, in the present case because the NV centre resides in the neutral instead of the negative charge state $\sim 30\%$ of the time[27]. In the neutral charge state no operation on the memory is possible and thus it cannot be flipped into the $|0\rangle$ state, which corresponds to a reduced signal.

*Multi-rotating frame.* For all our pulse sequences we use two MW frequencies, namely those resonant with the electron spin transition $m_S = 0 \leftrightarrow -1$ for the two $^{14}$N projections $m_I = +1, 0$ (that is, $|01\rangle \leftrightarrow |11\rangle$ and $|00\rangle \leftrightarrow |10\rangle$). Nuclear spin transitions on the $^{14}$N spin are only addressed in the $m_S = 0$ manifold (that is, $|00\rangle \leftrightarrow |01\rangle$) and therefore require only one RF (see Fig. 1b). RFs for $^{13}$C spins are swept over a large range around their bare Larmor frequency ($\approx 5.76\,$MHz).

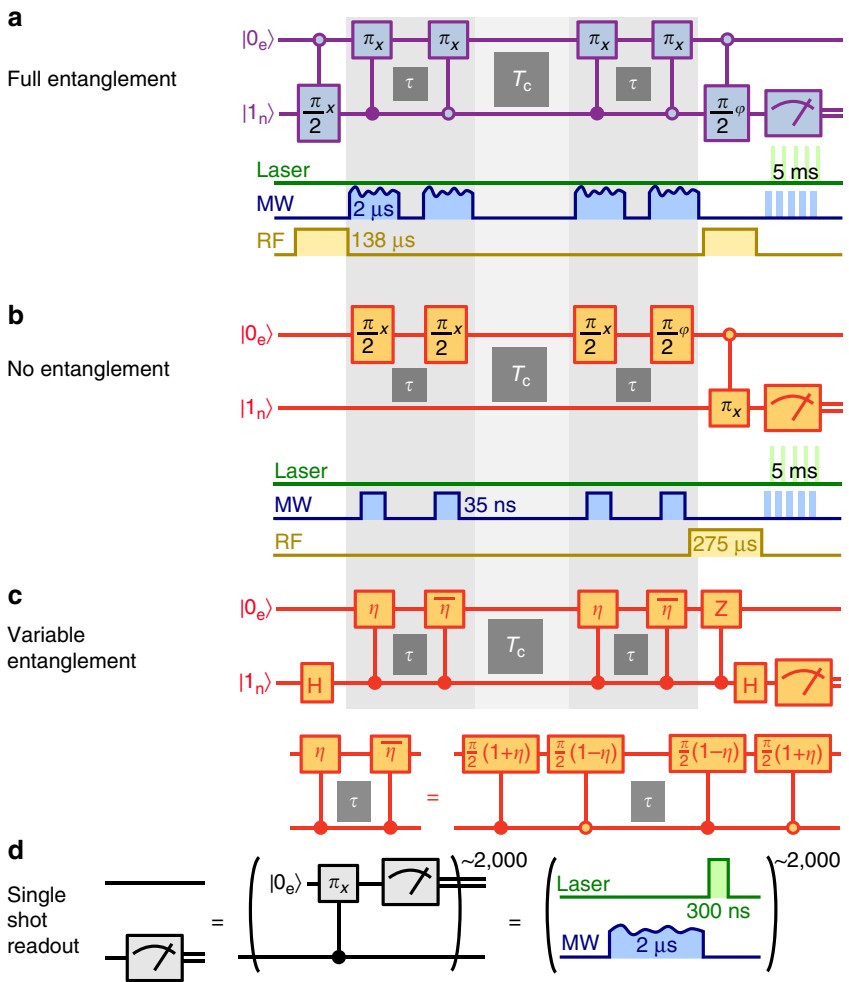

**Figure 6 | Correlation measurements with and without entanglement. (a–c)** Quantum wire diagrams for sensing with full quantum storage, classical storage and partial quantum storage, respectively. Qubit rotation operators $R_{x,y,z}(\theta)$[16] are abbreviated as $\theta_{x,y,z}$, for example, $\pi_x$ is a $\pi$ rotation around the x-axis of the respective rotating frame. Open (filled) circles indicate operation of the respective gate on the target qubit conditional to control qubit states $|0\rangle$ ($|1\rangle$). Experimental realizations via free evolution periods and laser, MW and RF pulses are sketched accordingly. **(d)** Quantum wire diagram and schematized drawing for SSR. The MW pulse used for the realization of the $C_nROT_e$-gate, here is illustrated by an amplitude modulated signal of duration 2 μs, which in our case is synthesized by the optimal control platform DYNAMO[26,41].

The resulting Hamiltonian in the respective multi-rotating frame is

$$H = \frac{1}{2}\begin{bmatrix} 0 & \Omega_N & \Omega_{0e} & 0 \\ \Omega_N & 2\delta_{0N} & 0 & \Omega_{1e} \\ \Omega_{0e} & 0 & 2\delta_{0e} & 0 \\ 0 & \Omega_{1e} & 0 & 2(\delta_{1e}+\delta_{0N}) \end{bmatrix}. \qquad (12)$$

Here, $\delta_{0N}$ is the RF detuning from the $^{14}N$ resonance in the $m_S = 0$ subspace, and $\delta_{0e}$ and $\delta_{1e}$ are the MW detunings from the electron spin resonance for $m_I = 0$ and $+1$ of the $^{14}N$ spin, respectively. We assume that all three mentioned transitions are addressed via their individual driving frequencies without crosstalk on the respective other transitions. In equation 12 the $\Omega$'s are the Rabi frequencies corresponding to the driving field strengths.

*Microwave pulses.* Almost all MW- and RF-pulses used in our experiments are controlled gates. All RF pulses acting on nuclear spins are realized by rectangular or Gaussian time-domain pulses, which is sufficient for selective addressing of the individual transitions without crosstalk on other transitions. For MW pulses acting on the electron spin, however, the spectrum is much more crowded and magnetic field drifts lead to broader resonance lines than for nuclear spins[28,43]. Hence, we optimize the $C_nROT_e$-gates, which are applied in the correlation sequence and the SSR measurements for robustness against magnetic field drift and fluctuation of the MW field strength and to avoid spectral crosstalk[41]. While a $C_nROT_e$-gate might be implemented by a rectangular time-domain pulse of Rabi frequency $\Omega \approx 1.25$ MHz and duration $t_p \approx 0.4$ μs, our pulse has modulated amplitude and phase and takes $t_p = 2$ μs (see Fig. 6).

During our novel correlation sequence, we apply four of these pulses sequentially. A full correlation sequence is simulated in Fig. 7 for $\tau = 0$, $T_c = 0$ and varying detuning. The comparably long duration of these pulses leads to phase accumulation on the memory spin even for sensing time $\tau = 0$ as illustrated in

Fig. 7b. Apparently, over a spectral range of $\sim 0.5$ MHz the quantum state of the memory spin shows almost full phase contrast and hence good fidelity. We introduce an effective sensing time $\tau_{eff} = \tau_0 + \tau$, where $\tau_0$ accommodates for phase accumulation during $C_nROT_e$-gate operations. The value of $\tau_0$ is estimated as follows.

When the magnetic field stays constant during $T_c$ also these finite length pulses do not lead to total phase accumulation. Hence, we consider a nuclear spin with hyperfine coupling $A_{zz} = 500$ kHz, which is flipped in $T_c$. During the first two $C_nROT_e$-gates it may be in state $\uparrow$ exhibiting an initial shift of 250 kHz. Thus it exhibits a shift of $-250$ kHz during the second two $C_nROT_e$-gates. According to our pulse analysis displayed in Fig. 7b this leads to a total phase accumulation of $\Delta\phi = 2\pi A_{zz}\tau_0 \approx 2\pi$. Hence, we estimate $\tau_0 \approx 2$ μs.

**Sensor and memory.** *Correlation measurement in general.* We use the electron spin as a sensor for magnetic fields and the nitrogen nuclear spin as an insensitive storage qubit. In the manuscript, the quantum wire diagram for the correlation measurement sequence with variable entanglement is displayed (see Fig. 1c orange part and Fig. 6c). However, for the two extreme cases of no and full entanglement, the diagrams simplify as described below (see Fig. 6a,b). For the most general description of our measurement sequence we have displayed the upper quantum wire diagram in Fig. 1c using Hadamard gates. Note that spin manipulations are usually rotation gates, for example, Hadamard gates become $\pi/2$-pulses and controlled Z-gates are implemented via hyperfine interaction.

*No entanglement.* In the conventional sequence case, the quantum gates on the electron spin sensor are local, that is, $\pi/2$ rotations that are insensitive to the memory state (see Fig. 6b). Therefore, these local gates can be performed very fast, in our case in 35 ns as compared with 2 μs in the case of the corresponding

$C_nROT_e$-gates. In that particular case we omit the initial $\pi/2$-rotation of the memory (conditional on $|0\rangle$), and instead replace the final $\pi/2$ by a $\pi$ rotation on the memory (conditional on $|0\rangle$). Thus, sensing and intermediate storage is solely performed by the sensor and only finally the sensor population of $|0\rangle$ is correlated with the memory state $|0\rangle$, which is eventually readout. In essence, this describes a classical stimulated echo sequence as it is known from EPR and optics[21,22].

*Full entanglement.* For full entanglement, the quantum gates on the sensor are fully selective to either memory states $|0\rangle$ or $|1\rangle$ (see Fig. 6a). In addition, the final controlled $z$-rotation on the sensor is omitted. Initial and final controlled rotations on the memory do not necessarily need to be controlled operations. However, single-frequency RF pulses are controlled operations, and for truly local gates, two or more RF pulses would be needed. The initial controlled gate on the memory excludes cases where the electron spin was not properly initialized, so its application is justified. Assuming high-fidelity $C_nROT_e$-gates on the sensor,

realized via optimal control, the final controlled gate on the memory again eliminates only cases of wrong initial sensor states.

Below, we describe the entire sequence for the case of full entanglement (cf. Figs 1c and 6a). As initial step, we prepare the electron (sensor) and the $^{14}$N (memory) spins in state $|01\rangle$ and then create the superposition state $|\Psi_i\rangle = |0\rangle \otimes (|0\rangle + |1\rangle)$ on the memory via a RF $\pi/2$-pulse. The subsequent pair of full entangling and disentangling gates are each a $C_nROT_e$-gate separated by a free evolution time $\tau$. We realize the latter gates by amplitude and phase modulated MW pulses (see optimal control section above) which flip the electron spin in a frequency/nuclear-spin-state selective manner[26,44] (see Fig. 6a). While the first sensor ROT-gate acts in a frequency band corresponding to state $|1\rangle$ of the memory, the second acts in a shifted frequency band corresponding to memory state $|0\rangle$. During $\tau$ the entangled state accumulates a phase as $|\Psi_\tau\rangle = |00\rangle + e^{i\phi_1}|11\rangle$. The acquired phase is stored on the nuclear spin memory during

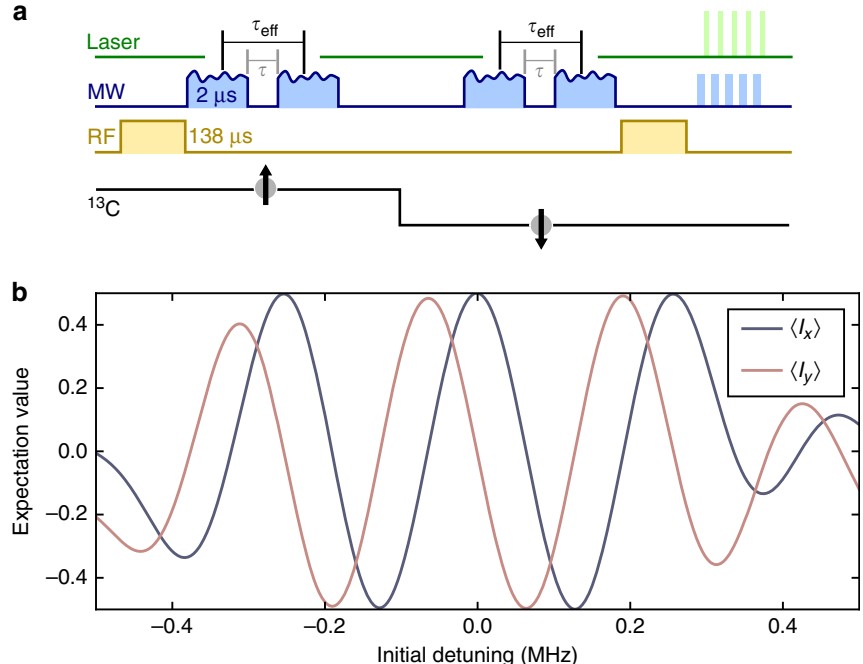

**Figure 7 | Effective sensing time $\tau_{eff}$. (a)** Illustration of effective sensing time $\tau_{eff}$ and actual $C_nROT_e$-gate separation $\tau$ during the sensing intervals of the correlation spectroscopy sequence (cf. Fig. 6) due to finite pulse length. A $^{13}$C sample spin (lower (black) line and arrow) adds an initial detuning to the sensor spin in the first $\tau_{eff}$, is flipped during $T_c$ and then adds the negative of the initial detuning to the sensor spin during the second $\tau_{eff}$. **(b)** Simulated phase of the memory spin after the second phase accumulation period as displayed in **a** for $\tau=0$ depending on the initial detuning created by the sample spin. The phase is depicted as $x$ and $y$ quadratures of the memory spin, $\langle I_x \rangle$ and $\langle I_y \rangle$, respectively. We attribute this phase to an accumulation time $\tau_0 = \tau_{eff} - \tau$ during the $C_nROT_e$-gate duration. Over a spectral range of approximately 0.5 MHz the expectation values show almost full contrast.

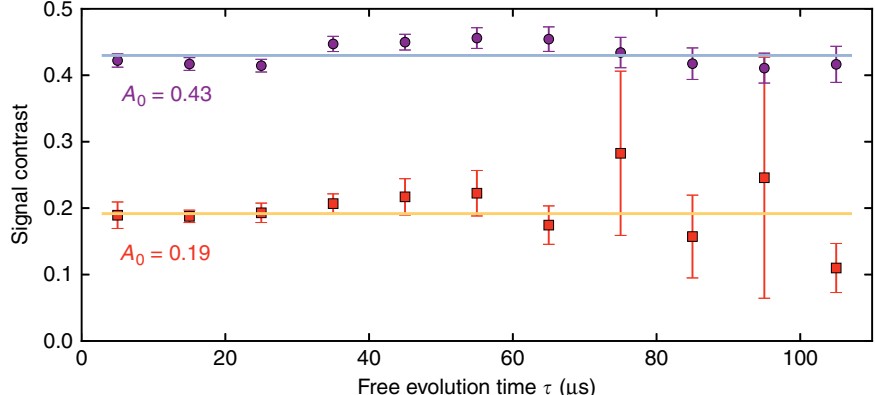

**Figure 8 | Quantity of stored information for conventional and enhanced method.** Analysis of fits to Fig. 2a,b, respectively. The initial signal contrast $A_0$ (that is, twice the initial amplitude of the decaying oscillation) is plotted for increasing sensing time $\tau$ and for enhanced and conventional measurement sequence (top and bottom, respectively). The horizontal lines are weighted averages of the signal contrast for both cases and their values are given. As these values represent the case of zero correlation time $T_c = 0$ they are analogous to the spin echo measurement displayed in Fig. 1d. Please note, that the error bars give standard errors taken from the used least square fit.

$T_c |\Psi_T\rangle = |1\rangle \otimes (|0\rangle + e^{i\phi_1}|1\rangle)$. The stored phase can be affected by a potential detuning $\Delta\omega$ leading to $|1\rangle \otimes (|0\rangle + e^{i(\phi_1 + \Delta\omega T_c)}|1\rangle)$. During $T_c$ either no operation is performed (see Fig. 2) or RF pulses resonant with $^{13}C$ sample spins are applied for the experiments displayed in Figs 3 and 4. These pulses are mainly rectangular time-domain pulses either of variable length (Fig. 3c,d) or corresponding to the length of a $\pi$-pulse (Figs 3a,b and 4). The spectra shown in Fig. 4a,b are recorded utilizing Gaussian time-domain pulses.

The final part of the correlation sequence again consists of the former pair of $C_nROT_e$-gates. Note that the special order of the two $|0\rangle$- and $|1\rangle$-controlled gates creates a spin echo or refocusing of the phase. The first gate creates $|\Psi_{\tau T_c}\rangle = |10\rangle + e^{i(\phi_1 + \Delta\omega T_c)}|01\rangle$, which is not $\Phi$-type Bell state as $|\Psi_\tau\rangle$, but a $\Psi$-type Bell state. Therefore, $|\Psi_{\tau T_c}\rangle$ then evolves into $|\Psi_{\tau T_c \tau}\rangle = (e^{i\phi_2}|10\rangle + e^{i(\phi_1 + \Delta\omega T_c)}|01\rangle)$ and is converted into

$$|\Psi_f\rangle = |0\rangle \otimes (|0\rangle + e^{i(\Delta\phi + \Delta\omega T_c)}|1\rangle) \tag{13}$$

by the last gate. Any changes in local magnetic field during correlation time $T_c$ lead to a phase difference $\Delta\phi$ which remains on the memory qubit and constitutes our metrology information.

In equation 13 we have distinguished $\Delta\phi$ from $\Delta\omega T_c$ where the first one represents difference of phase accumulation during both $\tau$ intervals, and the second one is phase accumulation due to detuning during $T_c$. The total detuning $\Delta\omega$ amounts to $2\pi(\delta_{0e} - \delta_{1e} - \delta_{0N})$. We set $\Delta\omega$ to zero, however, it slowly deviates from zero over time by much less than 1 kHz. Effects of that drift become apparent for instance in Fig. 3b of the main paper, where the background drifts instead of being constant over time. The drift may be caused by magnetic field variations or changes of the hyperfine coupling. If that drift causes any problems, a proper refocusing $\pi$-pulse on the memory spin can be applied. Minor phase offsets might be acquired during $\tau$ and during the initial and final RF pulses, which then alters $\Delta\phi$. However, because $\tau$ is much smaller than $T_c$ the latter effect is even smaller.

Instead of performing a phase estimation-like sequence as displayed in Fig. 6a, one can also utilize SWAP-gates. In that case one performs the first phase accumulation $\tau$ only on the sensor utilizing purely local gates[12]. Subsequently, the phase is swapped and kept on the memory during $T_c$. It is retrieved before the second sensing time $\tau$ via another SWAP-gate. Finally, the memory is correlated with the phase of the sensor. However, this is more costly in terms of RF pulses and interference. In fact, the latter fictitious sequence comprises at least a $3\pi$ flip angle of the memory, compared with only $\pi$ in the case of the phase estimation or conventional sequence, namely one for each of the two SWAP-gates and one for the final correlation of sensor and memory state for readout. Hence, it is less time and energy efficient. The latter fact is vital, since control of nuclear spins takes longer even at much higher driving field strengths due to their much smaller magnetic moment. In our case, for preparation and readout of the memory one $\pi$ rotation (that is, twice $\pi/2$) is necessary, which involves much time and strong RF fields (in the present case 138 μs and $\approx 2.4$ mT corresponding to $\approx 20$ W for our current setting, respectively). The SWAP-gate based approach requires significantly more resources. In total three $\pi$ pulses on the memory would be required (instead of one), the same amount of $C_nROT_e$-gates and additional operations on the sensor. Hence, for the current experimental setting 275 μs more time and more energy would be required. Interestingly, our novel sensing sequence requires a comparable quantity of gates, time and energy to the conventional sequence (see Fig. 6a,b). In addition, the storage and retrieval pulses on the memory act during the correlation time and therefore might interfere with what is going on during this time (for example, dynamics of interest or control of other spins).

Comparing conventional and enhanced correlation spectroscopy methods, we find that despite almost equal duration of both sequences, the enhanced method is slightly more complex because $C_nROT_e$-gates instead of local gates on the sensor are involved. The applied optimal control $C_nROT_e$-gates seem to cope well with the increased complexity. In fact, the signal contrast shows indeed the expected factor of two compared with the conventional method (see Figs 2a,b and 8). As the total durations ($> 5$ ms) of conventional and enhanced method differ only by 8 μs the doubled contrast directly leads to a doubled signal-to-noise ratio per unit time.

*Variable entanglement.* Variable entanglement between sensor and memory is created after initializing the system into $|\Psi_i\rangle = |0\rangle \otimes (|0\rangle + |1\rangle)$. The $\eta$-gate has the following effect (see Fig. 6c). The sensor qubit is rotated around $x$ by an angle $\pi/2$ to create an equal superposition state; this part is not entangling. Then, for memory state $|0\rangle$ the sensor qubit is rotated by an angle $-\eta \cdot \pi/2$ and for $|1\rangle$ it is rotated by an angle $+\eta \cdot \pi/2$, with $0 \leq \eta \leq 1$. The latter part is dependent on the memory state and therefore creates entanglement. The $\bar{\eta}$-gate is designed such that for no phase accumulation during $\tau$ the sensor state is finally $|1\rangle$ regardless of the memory state. As illustrated in Fig. 6c, the latter gates can be realized by multiple, nuclear spin state selective rotations of the electron spin around $x$ by the specified angles $(1 \pm \eta)\pi/2$. For the given initial state $|\Psi_i\rangle$ we can calculate the negativity $\mathcal{N}$ of the entangled state, where $\mathcal{N}$ is an applicable entanglement measure that ranges from $\mathcal{N} = 0$ for no entanglement to $\mathcal{N} = 0.5$ for a Bell state[24,25]. For varying $\eta$ we obtain

$$\mathcal{N} = \frac{\sin(\eta\frac{\pi}{2})}{2} \tag{14}$$

For the present case of variable entanglement we have to mind the hyperfine interaction which constitutes controlled $z$-rotations on the sensor spin and leads to

additional quantum correlations during $\tau$ and $T_c$. For our experiments with variable entanglement we choose the latter time intervals such that no additional correlations are created. Then we do the analogous experiments to those presented in Fig. 2. We fit the oscillations and from their amplitudes we deduce the amount of stored information (see Fig. 2d).

In addition to the experiment, we calculate the theoretically expected result for perfect sensor, memory, quantum gates and readout. For each degree of entanglement $\mathcal{N}$, we propagate the initial state $|\Psi_i\rangle\langle\Psi_i|$ through the entire sequence. During the correlation time $T_c$ we emulate decoherence of the sensor by deleting all coherences between different electron spin projections in the total density matrix $\rho$. Then we calculate the expectation value $Tr\{1 \otimes \sigma_z \cdot \rho\}$ (that is, we obtain the nuclear spin state populations). Finally, we average the latter result over the phase $\phi$, which is accumulated in the initial and final interval $\tau$, and we vary the phase difference $\Delta\phi$ of the final interval $\tau$ with respect to the initial interval $\tau$ and obtain

$$\langle Tr\{1 \otimes \sigma_z \cdot \rho\}\rangle(\Delta\phi) = \frac{1 + \cos^2(\eta \cdot \pi/2)}{2}\frac{\cos\Delta\phi}{2}$$
$$= f(\mathcal{N}) \cdot \frac{\cos\Delta\phi}{2} \tag{15}$$

$$f(\mathcal{N}) = \frac{1}{2} + 2\mathcal{N}^2 \tag{16}$$

Equation 15 results in an oscillation with $\Delta\phi$, just as observed in the experiment (cf. Fig. 2a,b). The prefactor in equation 15 can be expressed with the negativity $\mathcal{N}$ by substituting $\eta$ using equation 14 leading to equation 16. We have fit equation 16 to the measured signal contrasts for variable entanglement by scaling its function values (see Fig. 2d).

**Data availability.** The data supporting the findings of this study are available from the corresponding author on request.

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

## Acknowledgements

We thank Nabeel Aslam, Matthias Pfender, Nikolas Abt and Johannes Greiner for fruitful discussions. We acknowledge financial support by the German Science Foundation (SFB-TR 21, SPP1601, FOR1493) and the EU (SIQS, SQUTEC).

## Author contributions

All authors contributed extensively to the presented work. S.Z. and P.N. developed the initial strategy, S.-Y.L., S.Z., P.N. and J.W. planned the experiments and V.B. and T.S.-H. developed optimal control strategies. S.Z., T.R., I.J., T.W., S.W. and P.N. performed the experiments. S.Z., T.R., S.-Y.L., P.N. and J.W. wrote the manuscript in close contact with all authors. P.N. and J.W. supervised the project.

## Additional information

**Competing financial interests:** The authors declare no competing financial interests.

