## [Peer Review File · Nature Communications]

Reviewer #1 (Remarks to the Author):

The manuscript by Zaiser et al. reports a very nice experimental application of quantum techniques to improve the spectral resolution and sensitivity (signal contrast) of a hybrid system consisting of an electron spin sensing qubit and nuclear spin memory qubit in NV-diamond. The results are compelling: NV-NMR spectral resolution of about 200 Hz (albeit deep within the diamond), as well as a doubling in signal contrast for maximum sensing/memory qubit entanglement. The work is original and of great interest; and most of the manuscript's treatment of the relevant theory, experimental methodology, data, and conclusions are fine. There are some minor presentation issues that must be fixed: e.g., typos, ragged or incorrect English in many places, inconsistent capitalization, and font issues that make the identity of the nuclear spins (^{14}N and ^{13}C) not appear in the pdf file provided. The authors and Nature Communications editors need to do some careful editing of this manuscript, after which I strongly endorse its publication.

Reviewer #2 (Remarks to the Author):

In this manuscript the authors show how nuclear spin quantum memory can be used to improve by a factor of two the performance of an NV center as a magnetic field sensor. They use the correlation spectroscopy measurement scheme, which is well known to result in significant improvement in frequency resolution; the factor 2 sensitivity improvement in this work comes from the entanglement between the "sensor" electron qubit and the "memory" nuclear spin, which gets around the loss in signal contrast due to the sensor qubit coherence decay during the correlation time T_c . The authors test their scheme by demonstrating readout of a fictitious signal, introduced as a pulse phase shift, and then they use their scheme to detect the presence of nuclear spins in the diamond lattice, at various hyperfine coupling strengths.

In my opinion, this manuscript presents a useful metrological tool, which will find a number of applications in the field, but the text is in poor shape, and many revisions and clarifications are needed, as detailed below.

- 1) The authors claim that their method "makes the most efficient use of resources at hand" and that their algorithm makes "best use of sensing qubit and quantum memory". The manuscript does not substantiate these claims, indeed "most efficient" and "best" are not defined (frequency sensitivity, signal-to-noise, etc). These claims should be either removed or proven.
- 2) In eq. (1b) the average over ϕ gives a factor of 1/2 in the second term in the numerator, but the first term is still 1, which is inconsistent with the next line. The authors should correct the math or explain in more detail.
- 3) In fig. 2c the decay constant T_T displays exponential dependence on the free evolution time τ , with a time constant of about 30 μs . Why? Where does this time scale come from? It is much less than the electron spin T_2 time (400 μs).
- 4) In several places the authors refer to fig. S2c, which does not exist.
- 5) There are many places in the manuscript that need careful re-wording, such as "for enhanced respectively conventional measurement", "for we have $A_{zz}=\dots$ ", etc.
- 6) The authors use a strange notation where the phase appears as a superscript in an equation, I

suggest they use the usual exponential notation instead.

7) The authors claim that in fig. 3b they "can discriminate between the spin flipping up or down". They should explain in more detail how this can be seen from the data in the figure.

8) In Fig. 3c,d the authors chose τ to be 3.5 μ s and 0 μ s. Why? They mention that "effective τ is longer" - how much longer, and how does this affect the measurement? This should be quantified, and more information on the CROT gates etc may be useful here.

9) How do the authors define their "signal contrast" (fig. 1d and 2d) and "memory qubit signal" (fig. 3c) in terms of NV fluorescence readout? In fig. 3c, for example, this seems to be centered at 0.6 - why not 0.5? And why is the signal oscillation amplitude approx. 0.15? This is likely related to the amplitude of the peak in fig. 4a, but why is there an offset?

10) On a related note, the authors claim that "the addition of the memory did not show any disadvantages". Yet they do not quantify the time resources needed to implement the CROT gates (and the resulting reduction in signal-to-noise, given a fixed averaging time), and there is no mention of the fidelity of their gates. This information should be given in the manuscript. Indeed the ideal improvement offered by the authors' algorithm is a factor of 2 in sensitivity, but this will be degraded by imperfect gates and the extra time needed for the gates, so this should be quantified. In the "Full entanglement" subsection of "Methods" the authors claim that the 3π flip of the memory is "less time and energy efficient" than a single π flip, so clearly this is an issue.

11) The first sentence of the methods section refers to "a concentration of 0.2%" - this is concentration of what? The entire text should be carefully proof-read, as there are many such omissions and typos.

12) The authors mention that for spins the Hadamard gates are $\pi/2$ pulses, etc, yet they mix the notation in their quantum wire diagrams.

13) The following sentence in the caption of fig. 3 makes no sense: "For initialized spin, also the phase-shifted signal in panel b shows a signal"

14) The authors should be careful with referencing related work, for example I suggest at least citing Phys. Rev. X 5, 011001 (2015).

Response to referee comments on: 'Enhancing quantum sensing sensitivity and spectral resolution by a quantum memory'

April 24, 2016

1 General remarks

We thank all the referees and the editor for evaluating and considering our manuscript for potential publication. We are delighted about the overall positive remarks and we have answered all criticism raised by the referees in detail. We have revised the manuscript along these lines. More specifically, we have added requested details about the experiment, have added two figures to the methods section, have made less strong claims in some places and gave more profound reasons for the latter. Below we have given point by point answers to all comments, issues and questions raised by the referees. Furthermore, we have added a manuscript version with all major changes and additions highlighted in red.

Reviewers' comments:

Reviewer #1 (Remarks to the Author):

The manuscript by Zaiser et al. reports a very nice experimental application of quantum techniques to improve the spectral resolution and sensitivity (signal contrast) of a hybrid system consisting of an electron spin sensing qubit and nuclear spin memory qubit in NV-diamond. The results are compelling: NV-NMR spectral resolution of about 200 Hz (albeit deep within the diamond), as well as a doubling in signal contrast for maximum sensing/memory qubit entanglement. The work is original and of great interest; and most of the manuscript's treatment of the relevant theory, experimental methodology, data, and conclusions are fine. There are some minor presentation issues that must be fixed: e.g., typos, ragged or incorrect English in many places, inconsistent capitalization, and font issues that make the identity of the nuclear spins (^{14}N and ^{13}C) not appear in the pdf file provided. The authors and Nature Communications editors need to do some careful editing of this manuscript, after which I strongly endorse its publication.

We thank the referee for the evaluation of our work. In the revised manuscript we have added more experimental details, have refined our conclusions and claims. We also improved grammar, spelling and display of figures and text, to increase readability and clarity of our text.

Reviewer #2 (Remarks to the Author):

In this manuscript the authors show how nuclear spin quantum memory can be used to improve by a factor of two the performance of an NV center as a magnetic field sensor. They use the correlation spectroscopy measurement scheme, which is well known to result in significant improvement in frequency resolution; the factor 2 sensitivity improvement in this work comes from the entanglement between the "sensor" electron qubit and the "memory" nuclear spin, which gets around the loss in signal contrast due to the sensor qubit coherence decay during the correlation time T_c . The authors test their scheme by demonstrating readout of a fictitious signal, introduced

as a pulse phase shift, and then they use their scheme to detect the presence of nuclear spins in the diamond lattice, at various hyperfine coupling strengths.

In my opinion, this manuscript presents a useful metrological tool, which will find a number of applications in the field, but the text is in poor shape, and many revisions and clarifications are needed, as detailed below.

We thank the referee for the evaluation of our manuscript. We have revised our manuscript along the line of the raised issues given below.

- 1) The authors claim that their method "makes the most efficient use of resources at hand" and that their algorithm makes "best use of sensing qubit and quantum memory". The manuscript does not substantiate these claims, indeed "most efficient" and "best" are not defined (frequency sensitivity, signal-to-noise, etc). These claims should be either removed or proven.

We have removed all claims like "most efficient" and "best" use from the manuscript. Instead, we state precisely why our method is very efficient in certain aspects. First, we show that entanglement increases the signal by a factor of two for negligibly longer measurement time as compared to a standard measurement without quantum state storage on the memory. Second, given a single sensor and a single memory there is no way in storing more information on the memory than the full sensors quantum state. Improvements are envisioned for more memory qubits as stated in the conclusion. Third, we efficiently implement the quantum state storage via entanglement during sensing in a phase estimation like procedure which does not require SWAP gates. We explain that SWAP gates require a lot of RF power and comparably long time and are therefore less efficient. We have added lots of details about the resources used in our experimental sequences in order to judge our much more moderate claims. We do not exclude potential further improvements.

- 2) In eq. (1b) the average over ϕ gives a factor of 1/2 in the second term in the numerator, but the first term is still 1, which is inconsistent with the next line. The authors should correct the math or explain in more detail.

We have corrected eq. (1) such that the factor of 1/2 is now correctly displayed. In addition, we have added another detail concerning the constant shift of the central value of the oscillation from 0.5, which is typically observed in the experiment but was not included in the formula. The latter shift is due to different observable charge states of the NV center and is now explained in the text.

- 3) In fig. 2c the decay constant T_T displays exponential dependence on the free evolution time τ , with a time constant of about 30 us. Why? Where does this time scale come from? It is much less than the electron spin T_2 time (400 us).

From the questions raised in this point we deduce that we gave an improper explanation of what we have displayed in figure 2c. The sensor spin's coherence is expected to decay within time $T_2 = 2\tau$ for a correlation time $T_c = 0$ when increasing the sensing time τ as mentioned by the referee. In contrast here we have done the following. The data in figure 1a,b is analyzed such that for each τ we have extracted and plotted the signal decay time T_T , i.e. the correlation time T_c when the signal has decayed to 1/e. Therefore, figure 2c should be read in the following way. For $\tau = 0$ we expect the signal to decay to 1/e of its initial value after $T_c \approx 5$ ms (according to the fit). For $\tau = T_\tau \approx 30 \mu\text{s}$, however, we expect the signal to decay to 1/e of its initial value already after $T_c \approx 5$ ms/e.

We have rephrased the according paragraph.

- 4) In several places the authors refer to fig. S2c, which does not exist.

We intended to refer to fig. 6b (previously fig. 6c) instead. We have changed the manuscript accordingly.

- 5) There are many places in the manuscript that need careful re-wording, such as "for enhanced respectively conventional measurement", "for we have $A_{zz} = \dots$ ", etc.

We have carefully checked the manuscript to improve the readability and clarity of our text.

- 6) The authors use a strange notation where the phase appears as a superscript in an equation, I suggest they use the usual exponential notation instead.
 There were two unconventional notations with phase ϕ in superscript, eqs. (1b) and (7). In both cases ϕ was placed next to an overbar to indicate averaging over angle ϕ . We have removed the superscript and put a proper explanation in the text. In addition it seems that in some places equations were not properly displayed.
- 7) The authors claims that in fig. 3b they "can discriminate between the spin flipping up or down". They should explain in more detail how this can be seen from the data in the figure. We have revised the corresponding paragraph also with the help of the present equations to give a comprehensible explanation.
- 8) In Fig. 3c,d the authors chose τ to be 3.5 us and 0 us. Why? They mention that "effective τ is longer" - how much longer, and how does this affect the measurement? This should be quantified, and more information on the CROT gates etc may be useful here.
 We have revised the paragraph and we have given details about the $C_n\text{ROT}_e$ -gates in the appendix along with a figure explaining the meaning of τ_{eff} .
- 9) How do the authors define their "signal contrast" (fig. 1d and 2d) and "memory qubit signal" (fig. 3c) in terms of NV fluorescence readout? In fig. 3c, for example, this seems to be centered at 0.6 - why not 0.5? And why is the signal oscillation amplitude approx. 0.15? This is likely related to the amplitude of the peak in fig. 4a, but why is there an offset?
 We have added a full section about our readout technique and the appearance of the signal in the appendix. In essence, we perform a kind of quantum logic readout, where the memory spin's population is finally either correlated to its own or the sensor's quantum state (depending on the respective measurement sequence). The correlation with the memory's phase is established via a local $\pi/2$ -pulse whereas the correlation with the sensor's population is accomplished by a CNOT gate. Finally, we perform single shot readout of the memory spin, which yields "0" or "1" as a result and information about the spin state prior to the next measurement run. Hence, whenever we talk about signal, this refers to the probability of detecting the memory spin in state $|1\rangle$ given that it was properly initialized into state $|1\rangle$ at the beginning of the sequence. Whenever we talk about signal contrast, we have displayed the difference of two signals of the latter kind. The signal contrast is influenced by readout and initialization fidelity as well as decoherence and dissipation. Further the mentioned offset from 0.5 to ≈ 0.6 is due to the occurrence of the neutral instead of the negative charge state in $\approx 30\%$ of all experimental runs.
- 10) On a related note, the authors claim that "the addition of the memory did not show any disadvantages". Yet they do not quantify the time resources needed to implement the CROT gates (and the resulting reduction in signal-to-noise, given a fixed averaging time), and there is no mention of the fidelity of their gates. This information should be given in the manuscript. Indeed the ideal improvement offered by the authors' algorithm is a factor of 2 in sensitivity, but this will be degraded by imperfect gates and the extra time needed for the gates, so this should be quantified. In the "Full entanglement" subsection of "Methods" the authors claim that the 3π flip of the memory is "less time and energy efficient" than a single π flip, so clearly this is an issue.
 We have now given much more information about our experimental parameters to underline that our claims are indeed justified. First of all, we have indeed measured the signal improvement by a factor of two as displayed in fig. 2d. Second, we have now stated, that in our measurement sequence with a minimum duration of ≈ 5 ms we had to add less than $8\ \mu\text{s}$ to change from conventional to our improved measurement sequence, which is negligible. Hence, the signal to noise ratio is indeed twice larger for the enhanced as compared to the conventional sequence. Third, the discussion of the 3π RF pulses was aiming at a potential sequence containing swap gates, which would have made our sequence indeed less advantageous than it is now. However, we have avoided swap gates and were using entangled states during sensing, which we described as very efficient.

- 11) The first sentence of the methods section refers to "a concentration of 0.2%" - this is concentration of what? The entire text should be carefully proof-read, as there are many such omissions and typos.
We have proof-read the revised manuscript and in particular we have completed the sentence about the ^{13}C isotope concentration of the host diamond.
- 12) The authors mention that for spins the Hadamard gates are $\pi/2$ pulses, etc, yet they mix the notation in their quantum wire diagrams.
We have now written every quantum wire diagram with $\pi/2$ -pulses where applicable and have omitted Hadamard gates, except for the very first quantum wire diagram. There we want to describe the idea of our measurement in the most general way.
- 13) The following sentence in the caption of fig. 3 makes no sense: "For initialized spin, also the phase-shifted signal in panel b shows a signal"
We have carefully revised all figure captions.
- 14) The authors should be careful with referencing related work, for example I suggest at least citing Phys. Rev. X 5, 011001 (2015).
We have added several references, in particular the one mentioned here. We are sorry for having missed that one in the first place as it proposes a neat implementation of our quantum-sensor-memory-pair.

Reviewer #2 (Remarks to the Author):

The authors have addressed all the concerns raised in the original review. I recommend that the manuscript is published, although there are several typos remaining.

One final remark I have concerns the title: as it is, it implies that the spectral resolution is enhanced by using quantum memory, however that is not the case. The quantum memory does improve the sensitivity by a factor of 2 (eq. 6a and 6b, Fig. 8), but the spectral resolution is set by the correlation time T_c , which is independent of the entanglement between sensor and memory qubits. This can be seen in Fig. 2c. So in my opinion the manuscript title is a little misleading: the quantum memory in this work does not enhance spectral resolution.

Response to referee comments on: 'Enhancing quantum sensing sensitivity by a quantum memory'

May 30, 2016

1 General remarks

We thank all the referees and the editor for evaluating our manuscript again.

Reviewers' comments:

Reviewer #2 (Remarks to the Author):

The authors have addressed all the concerns raised in the original review. I recommend that the manuscript is published, although there are several typos remaining.

One final remark I have concerns the title: as it is, it implies that the spectral resolution is enhanced by using quantum memory, however that is not the case. The quantum memory does improve the sensitivity by a factor of 2 (eq. 6a and 6b, Fig. 8), but the spectral resolution is set by the correlation time T_c , which is independent of the entanglement between sensor and memory qubits. This can be seen in Fig. 2c. So in my opinion the manuscript title is a little misleading: the quantum memory in this work does not enhance spectral resolution.

We are glad to have addressed all main concerns. Regarding the controversial term "increased spectral resolution" in the title and in the manuscript, our intention was rather aiming at the increased spectral selectivity for coherent interactions to sample qubits. We find it hard to properly express this aspect in the title and we have therefor removed it. In the manuscript we have refined our conclusions:

In addition to the bare detection of proximal nuclear spin qubits we have also exploited the full storage of quantum information by demonstrating non-local quantum gates between the sensor-memory system and the proximal nuclear spins. Hence, we have increased the spectral selectivity of qubits with which our sensor-memory system can coherently interact beyond what is possible with the sensor alone or the sensor plus a classical memory.